# Improvement in sensitivity of radiochromic 3D dosimeter based on rigid polyurethane resin by incorporating tartrazine

Jin Dong Cho[1,2,3], Jaeman Son[1,4], Chang Heon Choi[1,2,4], Jin Sung Kim[3], Hong-Gyun Wu[1,2,4,5], Jong Min Park[1,2,4]*, Jung-in Kim[1,2,4]*

1 Department of Radiation Oncology, Seoul National University Hospital, Seoul, Republic of Korea,
2 Institute of Radiation Medicine, Seoul National University Medical Research Center, Seoul, Republic of Korea, 3 Department of Radiation Oncology, Yonsei University College of Medicine, Seoul, Republic of Korea, 4 Biomedical Research Institute, Seoul National University Hospital, Seoul, Republic of Korea, 5 Department of Radiation Oncology, Seoul National University College of Medicine, Seoul, Republic of Korea

* madangin@gmail.com (JIK); leodavinci@naver.com (JMP)

**Data Availability Statement:** All relevant data are within the manuscript.

**Funding:** This work was supported by a grant from the National R&D Program for Cancer Control,

## Abstract

We investigated the influence of incorporating tartrazine on the dose response characteristics of radiochromic 3D dosimeters based on polyurethane resin. We use three types of polyurethane resins with different Shore hardness values: 30 A, 50 A, and 80 D. PRESAGE dosimeters are fabricated with different chemical components and concentrations. Tartrazine (Yellow No. 5) helps incorporate a yellow dye to fabricate the dosimeter. Elemental composition is analyzed with the $Z_{eff}$. Three sets of six different PRESAGE dosimeters were fabricated to investigate the effects of incorporating yellow dye on the dose response characteristics of the dosimeter. The dose response curve was obtained by measuring the optical absorbance using a spectrometer and optical density using optical CT, respectively. The energy and dose rate dependences are evaluated for the dosimeter with the highest sensitivity. For the optical density measurement, significant sensitivity enhancements of 36.6% and 32.7% were achieved in polyurethane having a high Shore hardness of 80 D and 50 A by incorporating tartrazine, respectively. The same results were obtained in the optical absorbance measurements. The ratio of the $Z_{eff}$ of the dosimeter with 80 D Shore hardness to water was 1.49. The polyurethane radiochromic dosimeter with a Shore hardness of 80 D showed the highest sensitivity and energy and dose rate independence upon the incorporation of tartrazine.

## Introduction

Modern radiation treatment techniques, including three-dimensional (3D) conformal radiotherapy, intensity-modulated radiotherapy (IMRT), and volumetric-modulated arc therapy (VMAT), are being widely used in clinics for delivering highly conformal doses to the target volume while minimizing doses to organs at risk (OARs) [1,2]. Complex treatment delivery

Ministry of Health and Welfare, Republic of Korea (HA16C0025) to JK and the National Research Foundation of Korea (NRF) grant funded by the Korea government (MSIT) (0411-20190090) to HGW.

**Competing interests:** The authors have declared that no competing interests exist.

techniques such as IMRT and VMAT involve very steep dose gradients near the target volume and are therefore susceptible to errors in treatment delivery [3,4]. Thus, these sophisticated treatment techniques require a dosimeter with high dose sensitivity that can accurately measure dose distributions in three dimensions. In addition, pre-treatment patient-specific quality assurance (QA) for both IMRT and VMAT is highly recommended and routinely performed in clinics [5,6]. Patient-specific QA dosimetry systems in common use involve limited 2D dose measurement in practice. The gamma evaluation method was applied as a routine QA procedure to evaluate a planar dose distribution [7,8]. Several studies questioned the clinical relevance of 2D gamma evaluation. Stasi et al. and Nelms *et al.* demonstrated that no correlation was observed between the results of 2D gamma evaluation and clinically relevant patient dose errors for IMRT [9,10]. X Jin *et al.* showed there is no correlation between the percentage gamma passing rate and clinical dosimetric errors for both 2D and 3D pre-treatment VMAT dosimetric evaluation [11]. D. Rajasekaran *et al.* exhibited there is a lack of correlation or notable pattern for relation between planar 2D and volumetric 3D gamma analysis for VMAT plans [12]. Kim *et al.* demonstrated that no correlations were observed between the 2D and quasi-3D gamma passing rates for both IMRT and VMAT [13]. A critical need therefore arose for an accurate 3D dosimetry system that can provide a more comprehensive solution to the problem of verifying complex radiation treatment and performing more clinically relevant QA. As a result, 3D dosimeters were developed, which have a method of recording dose distributions in 3D. In the clinical practice, 3D dosimetry is not only performed by moving ionization chambers or silicon diodes or thermoluminescent detectors (TLD) in the water phantom but also chemical dosimetry systems such as polymer gels, radiochromic gels, and radiochromic plastics have been developed, recently [14,15]. These exhibit physical phenomena to radiation that changes their properties (e.g., optical absorption or scattering, X-ray absorption, NMR, or acoustic properties) [16,17]. Their changes are quantified and imaged by readout systems, which typically use magnetic resonance imaging (MRI) and optical CT systems. Polymer gel dosimeters consist of monomers dissolved in a viscous matrix that takes part in a polymerization reaction upon irradiation. Polymer gel dosimetry has been widely studied, and many researchers have demonstrated their limitations [18,19]. The use of Fricke gels as radiochromic gels for 3D dosimetry devices was proposed by Gore *et. al.*[20] In Fricke gels, when the Fricke solution is irradiated, ferrous ions ($Fe^{2+}$) are oxidized to ferric ions ($Fe^{3+}$) proportionally to the absorbed dose [21]. Fricke gel is relatively easy to prepare; however, degradation of the stored 3D dose distribution due to the diffusion of ferrous and ferric ions has been reported [22,23]. Both polymer and Fricke gels require an external casing to support them as they are not solid. In 2006, a radiochromic plastic material, "PRESAGE," was introduced as a novel 3D dosimetric system [24]. PRESAGE consists of a clear polyurethane matrix doped with a halogenated carbon radical initiator and radiation-sensitive reporter components. Upon exposure to radiation, free radicals generated from the hemolysis of the bonds between carbon and bromine lead to a color change caused by the radiolytic oxidation of the leuco dye [25]. This change in optical density (ΔOD) is linear with respect to the absorbed dose in the range of 0 to 100 Gy [20,24,26]. Some studies have reported that the energy and dose-rate dependence are negligible in the region from 145 kVp to 18 MV [24–27]. PRESAGE was proven to be capable as a 3D dosimetric system in several common clinical applications [28–30]. However, some other studies have reported substantial variations in dosimetric characteristics owing to different PRESAGE formulations. Mostaar et al. assessed the radiochromic responses of PRESAGE for various amounts of components used for fabrication [31]. They observed that when the concentration of the radical initiator was increased, PRESAGE dosimeter sensitivity increased while its stability decreased. Further, it was found that high concentrations of the radical initiator and leuco dye decreased the sensitivity of PRESAGE. Alqathami et al. investigated the

influence of three trihalomethane radical initiators on the sensitivity and stability of the PRES-AGE dosimeter [32]. They reported that iodoform incorporation in the composition of PRES-AGE enhanced the sensitivity of the dosimeter more than bromoform or chloroform. Moreover, Oldham demonstrated the wide variability in post-irradiation stability associated with relatively minor changes in polyurethane components [33]. Previous studies have reported that the dosimetric characteristics of PRESAGE dosimeters could be changed by varying the components in their composition. However, PRESAGE dosimeters have a disadvantage that changes absorbance by ultraviolet or visible light. For the reason, it should be stored in a dark environment to prevent from UV or visible light [32]. Similar issues appeared in the EBT film as well as the Presage dosimeter. To overcome this issue, EBT2 film was developed with the addition of yellow dye. Andres et al. performed a comparative dosimetric study for the EBT and EBT2 film (Gafchromic, International Specialty Products, Wayne, NJ, USA) to investigate the effect of yellow dye [34]. EBT2 has a yellow color owing to a dye incorporated in the active layer, which leads to different visible absorption spectra from that of the original EBT film. They reported that EBT2 was less sensitive to ambient light, probably owing to the yellow marker dye, which strongly absorbs blue light and this easy handling also helped to improve the film behavior due to ambient light resistance. The results show that the sensitivity for ambient light decreased owing to the addition of yellow dye in EBT2 film.

The yellow dyes could be tartrazine, eosin, quinoline yellow, metanil yellow, and particularly, useful dye among the tallow ydes is tartrazine [35]. Tartrazine is classified by azo compounds which are used as free radical initiators [36]. The sensitivity of PRESAGE dosimeter was related to free radical initiators [32]. We thought that tartrazine was influenced as free radical initiator for PRESAGE dosimeter. However, the effect of the tartrazine as free radical initiators on the dosimetric characteristics of PRESAGE dosimeter has not been investigated. Therefore, in this study, we investigate the influence of the tartrazine on dose response characteristics of the PRESAGE dosimeter. In addition, various polyurethanes are proposed to achieve optimal dosimeter properties with the addition of tartrazine as the yellow dye.

## Materials and methods

### Radiochromic plastic dosimeters and formulations

Radiochromic plastic dosimeters were fabricated with compositions of the commercial PRES-AGE dosimeter using three well-known components that included a transparent polyurethane plastic prepolymer mixture, leuco dye, and a radical initiator. In this study, we used three types of polyurethane resins with different Shore hardnesses: Clear Flex™ 30 at 30 A, Clear Flex™ 50 at 50 A, and Crystal Clear™ 200 at 80 D (Smooth-On, Easton, PA, USA). These polyurethane resins were supplied in two parts (Part A and Part B). Part A is an aliphatic diisocyanate and Part B is a polymer with hydroxyl functional groups [37]. Leucomalachite green (LMG, 98%, Aladdin Chemical Co. Ltd., China) was used as a reporter compound. LMG is well known as the most desirable leuco dye in the formulation of the commercial PRESAGE. Tetrabromo-methane ($CBr_4$, 98%, Aladdin Chemical Co. Ltd., China) was used as a radical initiator. Both dimethylsulfoxide (DMSO, extra pure, Daejung Chemical Co., Ltd., Korea) and acetone (guaranteed reagent, Daejung Chemicals & Metals Co., Ltd., Korea) were used to dissolve other components. Tartrazine (Yellow No. 5, Daejung Chemical Co., Ltd., Korea) was used to incorporate a yellow dye in dosimeter fabrication. The fabrication process involved the following steps: (i) $CBr_4$, LMG, solvents, and tartrazine were thoroughly mixed with the Part B compound; (ii) the Part A compound was then added and mixed with Part B with vigorous stirring; (iii) this final mixture was then poured into poly spectrometer cuvettes with the dimensions of $10 \times 10 \times 45$ mm$^3$ (Heuris Inc., Skillman, NJ, USA), and the filled cuvettes were

**Table 1. Formulations for radiochromic polyurethane dosimeters.** Percentages are by weight. Formulation names are based on formulation characteristics using tartrazine.

| Formulation | [a]CF30 | [b]CF30T | [c]CF50 | [d]CF50T | [e]CC200 | [f]CC200T |
|---|---|---|---|---|---|---|
| Polyurethane | Clear Flex 30 | Clear Flex 30 | Clear Flex 50 | Clear Flex 50 | Crystal Clear 200 | Crystal Clear 200 |
| | 46.16% Part A 41.54% Part B | 46.13% Part A 41.52% Part B | 29.23% Part A 58.47% Part B | 29.22% Part A 58.43% Part B | 46.16% Part A 41.54% Part B | 46.13% Part A 41.52% Part B |
| Leuco dye | 2.00% [g]LMG | | | | | |
| Initiator | 4.00% [h]$CBr_4$ | | | | | |
| Solvents | 2.00% [i]DMSO, 4.30% Acetone | | | | | |
| Yellow dye | | 0.05% Tartrazine | | 0.05% Tartrazine | | 0.05% Tartrazine |
| Shore hardness | 30 A | | 50 A | | 80 D | |

[a]CF30 = Clear Flex™ 30.

[b]CF30T = CF30 adding tartrazine.

[c]CF50 = Clear Flex™ 50.

[d]CF50T = CF50 adding tartrazine.

[e]CC200 = Crystal Clear™ 200.

[f]CC200T = CC200 adding tartrazine.

[g]LMG = Leucomalachite green.

[h]$CBr_4$ = Tetrabromomethane.

[i]DMSO = dimethylsulfoxide.

placed in a pressure pot (60 psi) for 48 h to minimize out-gassing. Three sets of six different PRESAGE dosimeters within a single batch were fabricated to investigate the effects of incorporating yellow dye on the sensitivity of the dosimeters. The chemical components and their concentration for each formulation are listed in Table 1.

In addition, these formulations were investigated by performing elemental analysis of the compounds. The effective atomic number ($Z_{eff}$) of each formulation was calculated using X-ray fluorescence (XRF) measurements (S8 Tiger, Brunker Co., Billerica, MA). This was then used for the elemental composition analysis. The atoms of the elements can be determined using the characteristic X-ray; in addition, the intensity of the X-ray fluorescence is proportional to the concentration of the element in the sample.

## Dosimeter irradiation

A custom-made acrylic phantom was fabricated to insert cuvettes for delivering a uniform dose to a total of three cuvettes at a time. The dimensions of the phantom were $10 \times 10 \times 14$ cm$^3$. Three cuvettes could be located at the center of the phantom, i.e., the center of each cuvette was located at a depth of 7 cm into the phantom, as shown in Fig 1(A).

The mold phantom with three inserted cuvettes was processed to acquire CT images using a Brilliance CT Big Bore™ (Philips, Cleveland, OH, USA) with an imaging slice thickness of 1 mm. Based on these CT images, IMRT plans using two opposed bilateral beams were calculated to deliver uniform doses of ±1.0% to all cuvettes at once. The IMRT plans were generated with 6 MV photon beams in the Eclipse™ system (Varian Medical Systems, Palo Alto, CA USA). Irradiation of the PRESAGE dosimeters was also carried out with a 6 MV linear accelerator (Varian Medical Systems, Palo Alto, CA, USA) according to the IMRT plan. Various radiation doses (0, 10, 20, 50, 80, 100, 150, 200, and 300 cGy) were delivered at a dose rate of 600 cGy/min for all fabricated dosimeters, as shown in Fig 2.

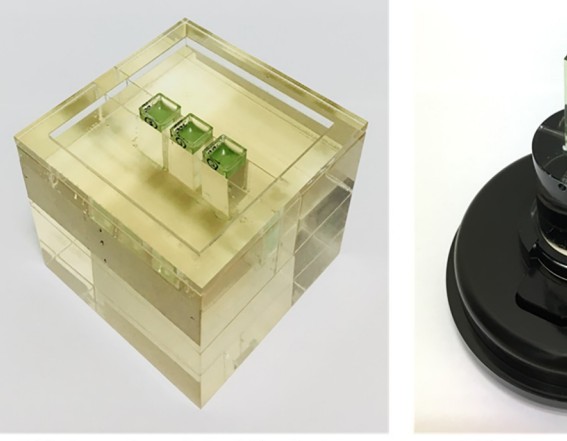

(a) Custom-made cuvette irradiation phantom          (b) Custom-made cuvette holder

**Fig 1.** (a) Custom-made cuvette irradiation phantom and (b) custom-made cuvette holder.

## Optical absorption measurements and optical CT scanning

All fabricated PRESAGE dosimeters were stored in a cold (4–6°C) and dark environment pre- and post-irradiation to avoid any absorbance change due to exposure to visible light [38]. For the LMG, the typical visible maximum absorption wavelength ($\lambda_{max}$) of its oxidized form (malachite green) is well-known as approximately 633 nm. An absorption spectrum, which relates to the absorbance as a function of wavelength, can be used to select the optimal wavelength for absorbance acquisitions in each sample reference. Absorption spectra were acquired to determine the $\lambda_{max}$ of the fabricated dosimeters with different absorbed doses (0, 10, 50, 100, 200, and 300 cGy) using the Eppendorf BioSpectrometer® (Eppendorf, NY, USA). A Xenon flash lamp can emit a broad spectrum (200–830 nm) in 1-nm intervals, but we used the 400–800 nm region in this study. The dimension of the cuvette shaft in the spectrometer was identical to that of the dosimeter cuvette.

After irradiation, a cone-beam optical CT scanner (Vista™ Optical CT Scanner, Modus Medical Devices Inc., Ontario, Canada) was used for acquisition of 3D images for optical density (OD) of cuvettes. This optical CT scanner had a diffused light source (light-emitting diode), and we used a bandpass filter with a central wavelength frequency of 633 nm. For each scan, a set of 512 light-intensity transmission projections (640 × 480 pixels each) were acquired over 360°. This resulted in the production of 256 × 256 × 256 elements with a reconstructed voxel resolution of 0.5 × 0.5 × 0.5 mm³. A shutter speed of 25.0 ms and a frame rate of 7.5 fps were used. The OD data for the reconstructed 3D images were acquired using MicroView™ software (Parallax Innovations, Ontario, Canada). The region of interest (ROI) at the center of a cuvette was defined as an 8 × 8 × 8 mm³ cubes and we reported ODs averaged over all voxels in the ROI. We designed a cuvette holder as another custom-made device to enable reproducible fixation and orientation of the cuvettes during scanning as shown in Fig 1(B). To minimize light-scattering artifacts, we used silicone oil (KF-54, Shin-Etsu Chemical Co. Ltd., Tokyo, Japan) and filled it into the aquarium during scanning. The oil was transparent and had a refractive index of 1.505, which was well-matched with those of fabricated dosimeters including cuvettes (refractive index 1.51 ± 0.01).

## Response curve and measurements of sensitivity

From the optical absorption with spectrometer, the dose sensitivity (Δabsorbance/(Gy·cm)) was defined as the slope of the dose response curve, which was the maximum Δabsorbance at $\lambda_{max}$ with absorbed doses. The change in absorbance (Δabsorbance) at $\lambda_{max}$ were obtained by

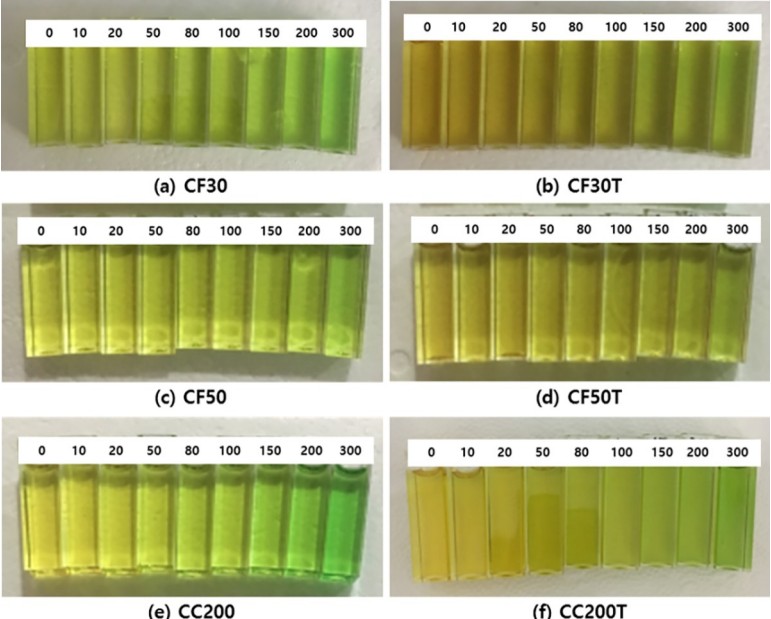

**Fig 2. Representative photographs of the fabricated dosimeter cuvettes with different polyurethanes incorporating tartrazine.** (a) The CF30 formulation refers to polyurethane Clear Flex™ 30 without tartrazine, (b) the CF30T formulation refers to polyurethane Clear Flex™ 30 with tartrazine, (c) the CF50 formulation refers to polyurethane Clear Flex™ 50 without tartrazine, (d) the CF50T formulation refers to polyurethane Clear Flex™ 50 with tartrazine, (e) the CC200 formulation refers to polyurethane Crystal Clear™ 200 without tartrazine, and (f) the CC200T formulation refers to polyurethane Crystal Clear™ 200 with tartrazine. Each cuvette was irradiated with various doses (0, 10, 20, 50, 80, 100, 150, 200, and 300 cGy).

subtracting the absorbance at $\lambda_{max}$ for the un-irradiated cuvette from that of the irradiated cuvettes. For 3D reconstructed images obtained from optical CT scanner and software, the dose sensitivity ($\Delta OD/(Gy \cdot cm)$) was defined as the slope of the dose response curve, which was the $\Delta OD$ with the absorbed doses. The $\Delta OD$ was obtained by subtracting the $OD_0$ for the un-irradiated cuvettes from OD of the irradiated cuvette. The dose response curve was obtained by plotting the $\Delta OD$ with optical CT scanner and maximum $\Delta$absorbance at $\lambda_{max}$ with spectrometer as a linear function of absorbed dose. The sensitivity enhancement was defined as the ratio between the slopes of the different fabricated dosimeters. Finally, the goodness of fit of the dose response curve plotted with the straight line was investigated with a coefficient of determination ($R^2$) for all fabricated dosimeters.

## Evaluation of dosimetric characteristics

Energy and dose rate dependence studies were conducted on the CC200T formulation irradiated to 100 cGy. The energy dependence studies were performed using flattening filtered beams (6, 10, 15 MV photon beams) and a flattening filter free beam (6 MV FFF photon beam). For a given energy level, the mean pixel values of each dosimeter were obtained and normalized to that of the 6 MV photon beam. Dose rate dependence studies were also performed using the 6 MV and 6 MV FFF photon beams. The dose rate range of the 6 MV photon beam is from 100 to 600 MU/min, and that of the 6 MV FFF photon beam is from 400 to 1400 MU/min, of which the latter is relatively high. The dose rates of the 6 MV and 6 MV FFF photon beams were spaced 100 MU/min and 200 MU/min apart, respectively. For each dose rate, the mean pixel values of each dosimeter were obtained and normalized to the median dose rates of 300 MU/min and 800 MU/min for the 6 MV and 6 MV FFF photon beams, respectively.

## Results

### Elemental composition analysis

From the XRF measurements, the $Z_{eff}$ value of the CC200 and CC200T formulations was calculated to be 11.1; in addition, other formulations showed the same value to be 10.5 in Table 2. The chemical compositions of all the fabricated dosimeters included C, H, O, N and Br. However, the fabricated dosimeters incorporating tartrazine contained a Na component. The ratio of the $Z_{eff}$ value of the CC200 and CC200T formulations to that of water was 1.49, and that of other formulations was 1.42.

### Response curve and sensitivity from absorption spectrum

Absorbance acquisitions were conducted at different times (0.5, 1, 2, 3, 6, and 16 h) post-irradiation with the spectrometer and optical CT scanner. The Δabsorbance were very stable within two hours post-irradiation [25]. These results are consistent with the OD measurements. The optical absorption spectra were obtained for all fabricated dosimeters, which were irradiated with various doses and showed different absorbance changes. As shown in Fig 3, the spectrum of the unexposed dosimeter had bands centered at about 425 nm and 630 nm except for the CF30T formulation, which had a band at 627 nm. The peak absorbance was not changed by the incorporation of tartrazine. However, for all formulations, the optical absorbance was significantly enhanced by its incorporation. The ratio of maximum optical absorbance at $\lambda_{max}$ between the dosimeters using the same polyurethane was the highest at low doses and the tendency was shown to decrease as the dose increased. After exposure, peak

**Table 2. Elemental composition analysis results of the XRF measurements for all formulations.**

| Formulation | [a]CF30 | [b]CF30T | [c]CF50 | [d]CF50T | [e]CC200 | [f]CC200T |
|---|---|---|---|---|---|---|
| [g]$Z_{eff}$ | 10.5 | 10.5 | 10.5 | 10.5 | 11.1 | 11.1 |
| [h]Ratio_water | 1.42 | 1.42 | 1.42 | 1.42 | 1.49 | 1.49 |
| [i]H | 9.8% | 9.8% | 9.7% | 9.8% | 9.4% | 9.5% |
| [j]C | 64.0% | 64.0% | 61.0% | 60.9% | 61.6% | 62.4% |
| [k]N | 2.8% | 2.7% | 3.5% | 3.2% | 5.3% | 5.4% |
| [l]O | 20.1% | 20.3% | 22.8% | 23.1% | 20.1% | 19.0% |
| [m]S | 0.3% | 0.2% | 0.3% | 0.3% | 0.4% | 0.4% |
| [n]Br | 2.9% | 2.9% | 2.6% | 2.6% | 3.2% | 3.2% |
| unknown | 0.1% | 0.1% | 0.1% | 0.1% | 0.1% | 0.1% |

[a]CF30 = Clear Flex™ 30.

[b]CF30T = CF30 adding tartrazine.

[c]CF50 = Clear Flex™ 50.

[d]CF50T = CF50 adding tartrazine.

[e]CC200 = Crystal Clear™ 200.

[f]CC200T = CC200 adding tartrazine.

[g]$Z_{eff}$ = Effective atomic number.

[h]Ratio_water = The ratio of the $Z_{eff}$ value of the formulations to that of water.

[i]H = Hydrogen.

[j]C = Carbon.

[k]N = Nitrogen.

[l]O = Oxygen.

[m]S = Sulfur.

[n]Br = Bromine.

absorption occurred at the maximum-absorption wavelengths (i.e., 627 nm and 630 nm) but a strong absorption at 425 nm remained. The Δabsorbance at $\lambda_{max}$ were obtained by subtracting the absorbance at $\lambda_{max}$ for the un-irradiated cuvette from that of the irradiated cuvettes. The averaged change in absorption spectra of all fabricated dosimeters for different doses are shown in Fig 3. The Δabsorbance at $\lambda_{max}$ increased for formulations with Clear Flex 50 (Shore hardness 50 A) and Crystal Clear 200 (Shore hardness 80 D) by incorporating tartrazine. On the contrary, the Δabsorbance at $\lambda_{max}$ decreased for the formulation with the Clear Flex 30 (Shore hardness 30 A) by incorporating tartrazine. The average maximum optical Δabsorbance at $\lambda_{max}$ through various irradiated doses was increased by 63.9%, 21.8%, and -38.3% for CC200T, CF50T, and CF30T as compared with CC200, CF50, and CF30, respectively.

The dose response curves are plotted with the maximum Δabsorbance values at $\lambda_{max}$ for each formulation, as shown in Fig 4. The dose sensitivity was significantly increased for CF50T and CC200T by the incorporation of tartrazine. The sensitivity enhancements were 69.2% and 12.6% for CF50T and CC200T as compared with CF50 and CC200, respectively. However, the dose sensitivity was significantly decreased for CF30T by incorporating tartrazine and the sensitivity enhancement was -9.5% as compared with CF30. The linear least squares method was used to provide the best line of fit for absorbance against dose.

For all formulations, very good coefficient of determination ($R^2 > 0.99$) for the dose response was exhibited over the applied radiation dose range. The $R^2$ value for CC200 and CC200T were 0.9980 and 0.9992, respectively. These results are summarized in Table 3.

## Response curve and sensitivity from optical CT scanner

The optimal wavelength for OD acquisitions was determined by the spectrophotometry of all fabricated dosimeters. However, the OD values were obtained using a central wavelength of 633 nm owing to the limitations of the optical CT scanner. The averaged dose response curves were plotted for each formulation as shown in Fig 5.

As with the optical absorbance measurement results, the dose sensitivity was significantly increased for CF50T and CC200T by incorporating tartrazine. The sensitivity enhancements were 36.6% and 32.7% for CF50T and CC200T as compared with CF50 and CC200, respectively. In particular, the formulation with Crystal Clear 200 having Shore hardness 80 D showed significant sensitivity enhancement by incorporating tartrazine. However, the dose sensitivity was also significantly decreased for CF30T by incorporating tartrazine and the sensitivity enhancement was -39.2% as compared with CF30. The averaged OD through various irradiated doses was increased to 53.5%, 12.4%, and -4.0% for CC200T, CF50T, and CF30T as compared with CC200, CF50, and CF30, respectively. The ratio of absolute OD values between the dosimeters using the same polyurethane was also the highest at low doses and the tendency was shown to decrease as the dose increased. By incorporating tartrazine, the absolute OD of all fabricated dosimeters was increased. The ratio between CC200 and CC200T was the largest among other formulations. This result is consistent with the most prominent tartrazine effect at CC200T having the highest dose sensitivity. The linear least squares method was also used to provide the best line of fit for absorbance against dose. A very good $R^2$ value ($> 0.99$) for the dose response was observed for all formulations with OD measurements. These results are summarized in Table 4.

## Dosimetric characteristics of CC200T formulation

The energy response of the fabricated dosimeters is shown in Fig 6. The difference for 10 and 15 MV photon beams was -0.73% and -0.35% compared to the 6 MV photon beam,

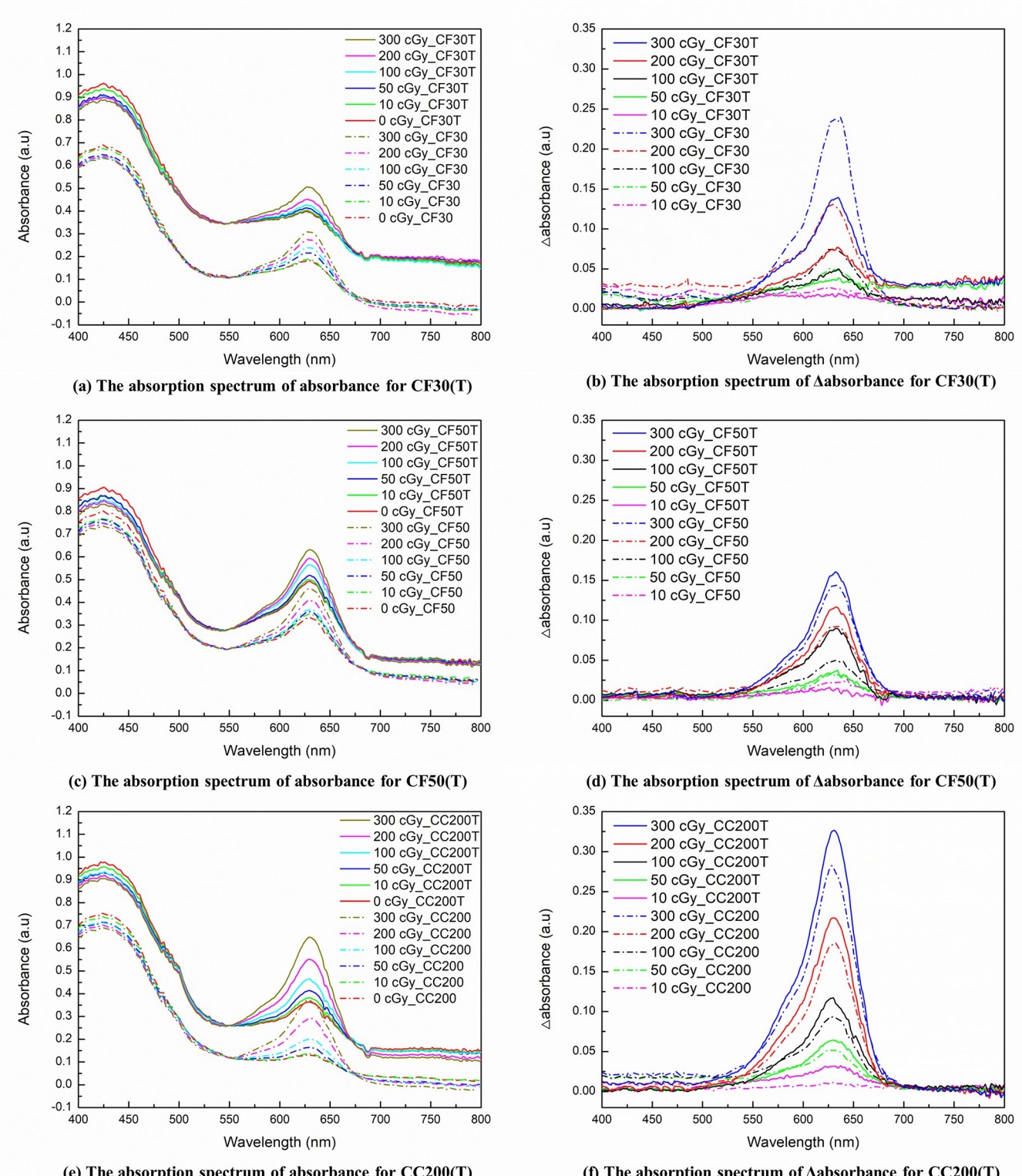

**(a) The absorption spectrum of absorbance for CF30(T)**

**(b) The absorption spectrum of Δabsorbance for CF30(T)**

**(c) The absorption spectrum of absorbance for CF50(T)**

**(d) The absorption spectrum of Δabsorbance for CF50(T)**

**(e) The absorption spectrum of absorbance for CC200(T)**

**(f) The absorption spectrum of Δabsorbance for CC200(T)**

**Fig 3. Representative optical absorbance plotted against the wavelength spectrum for the fabricated dosimeters with different irradiation doses and the wavelength spectrum of Δabsorbance acquired by subtracting the absorbance of the un-irradiated cuvette from that of the irradiated cuvettes.** (a) The absorption spectrum of absorbance for CF30 and CF30T formulations refer to polyurethane Clear Flex™ 30. (b) The absorption spectrum of Δabsorbance for CF30 and CF30T. (c) The absorption spectrum of absorbance for CF50 and CF50T formulations refer to polyurethane Clear Flex™ 50. (d) The absorption spectrum of Δabsorbance for CF50 and CF50T. (e) The absorption spectrum of absorbance for CC200 and CC200T formulations refer to polyurethane Crystal

Clear™ 200. (f) The absorption spectrum of Δabsorbance for CC200 and CC200T. Solid lines denote formulations incorporating tartrazine and dash-dotted lines denote formulations without tartrazine.

respectively. For the 6 FFF photon beam, the difference was -0.95%. In conclusion, the results obtained show no significant energy dependence for the given photon beams.

The dose rate dependence obtained for the fabricated dosimeters is presented in Fig 7. The maximum response difference for the 6 MV photon beam was observed at 400 MU/min. Further, the maximum response difference for the 6 MV FFF photon beam was observed at 600 MU/min. However, these differences were 1.6% and -1.7% compared to the median dose rate, respectively. Thus, the results obtained show no significant dose rate dependence for the dosimeters.

## Discussion

In this study, we investigated the influence of tartrazine as a yellow dye on the dose response characteristics of PRESAGE dosimeters with various polyurethanes using optical absorbance and optical density measurements. By incorporating tartrazine, the sensitivity was significantly increased for two kinds of polyurethane resins: Clear Flex 50 (Shore hardness 50 A) and Crystal Clear 200 (Shore hardness 80 D). The yellow dye such as tartrazine was added in EBT2 film developed to replace EBT film in 2009. The most notable feature of EBT2 film is the addition of a yellow marker dye in the active layer, which strongly absorbs blue light. It can be expected to reduce the effect of light exposure on the active component of the film and can also be used to normalize the response to small changes in the thickness of the film's active layer [34]. The spectrum of the unexposed EBT2 has bands centered at 420 nm, 585 nm, and 636 nm. After exposure, the peak absorptions occur at about 636 nm and 585 nm since the active component in the film reacts to form a blue-colored polymer. On the other hand, a strong absorption at 420 nm due to the yellow marker dye remains, although the peak absorbance is slightly increased. This increase is due to the contribution of the dye polymer resulting from

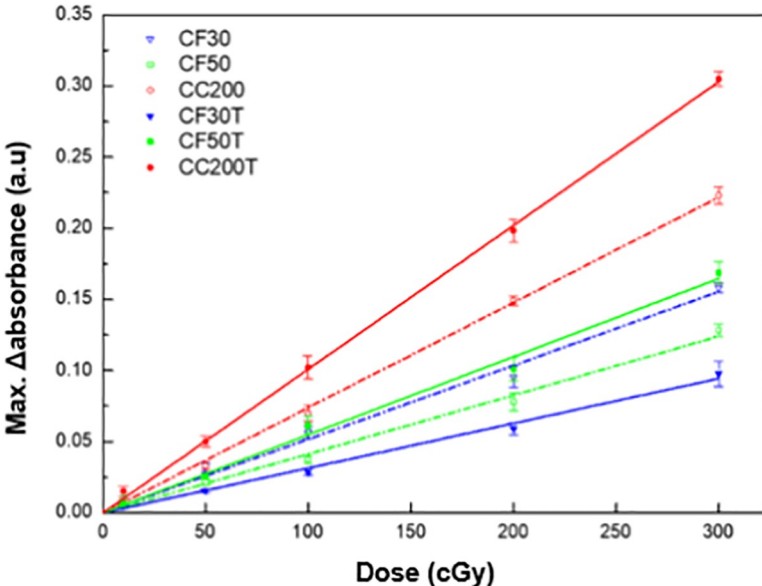

**Fig 4. Maximum Δabsorbance values at $\lambda_{max}$ for each formulation as a function of the absorbed dose.**

**Table 3. For optical absorption measurements, the maximum Δabsorbance at λ$_{max}$ was observed for all fabricated dosimeters.** The dose sensitivity was defined as the slope of the dose response curve, which was the maximum Δabsorbance at λ$_{max}$ with absorbed doses. The sensitivity enhancements were defined as the ratio between the slopes of the different fabricated dosimeters.

| Formulation | [a]CF30 | [b]CF30T | [c]CF50 | [d]CF50T | [e]CC200 | [f]CC200T |
|---|---|---|---|---|---|---|
| λ$_{max}$ (nm) | 630 | 627 | 630 | 630 | 630 | 630 |
| Sensitivity at λ$_{max}$ (Δabsorbance /(Gy·cm)) | 0.0189 | 0.0171 | 0.0271 | 0.0305 | 0.0425 | 0.0719 |
| Sensitivity enhancement | - | -9.5% | - | +12.6% | - | +69.2% |
| $R^2$ value at λ$_{max}$ | 0.9921 | 0.9954 | 0.9945 | 0.9964 | 0.9980 | 0.9992 |

[a]CF30 = Clear Flex™ 30.

[b]CF30T = CF30 adding tartrazine.

[c]CF50 = Clear Flex™ 50.

[d]CF50T = CF50 adding tartrazine.

[e]CC200 = Crystal Clear™ 200.

[f]CC200T = CC200 adding tartrazine.

irradiation. The secondary absorbance peak at 585 nm, characteristic of the dye polymer, has a tail on the low-wavelength side that extends below 400 nm and into the UV region. The effect of this tail is to increase the absorbance at 420 nm after exposure. It is well-known that tartrazine has a maximum absorbance of 426 nm. Thus, a yellow marker dye does not directly affect the sensitivity of the film. In this study, the spectrum of the unexposed dosimeters has bands centered at about 630 nm except for the CF30T formulation, which has a band centered at 627 nm. However, the peak absorbance did not change by incorporating tartrazine. There remains an open question in the current experiment as to whether tartrazine inclusion makes a dosimeter less sensitive to ambient light and thereby improves its characteristics.

Sensitivity was significantly increased for two kinds of polyurethane resins: Clear Flex 50 (Shore hardness 50 A) and Crystal Clear 200 (Shore hardness 80 D) by incorporating the yellow dye. On the contrary, the dosimetric characteristics of the dosimeter using the Clear Flex 30 polyurethane (Shore hardness 30 A) were degraded by incorporating yellow dye. Juang

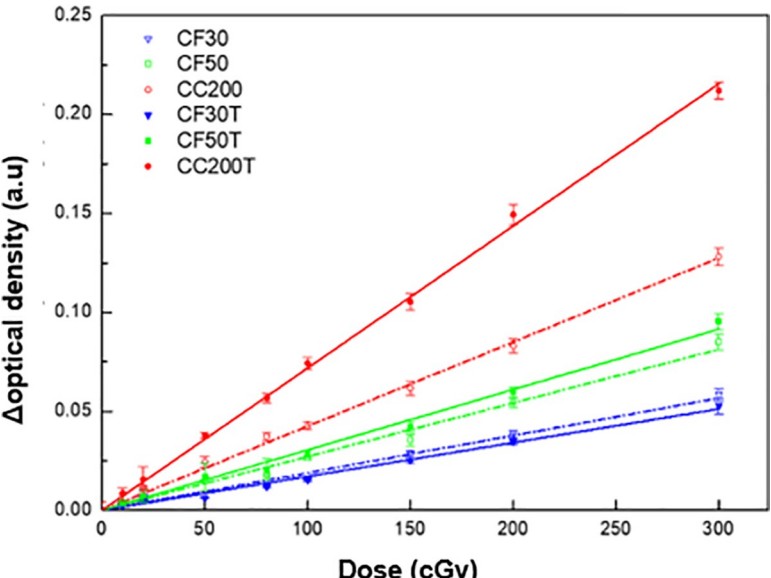

**Fig 5. Optical density changes (ΔOD) for each formulation as a function of the absorbed dose.**

**Table 4. For optical density measurements, the dose sensitivity was defined as the slope of the dose response curve.** The sensitivity enhancements were defined as the ratio between the slopes of the different fabricated dosimeters.

| Formulation | [a]CF30 | [b]CF30T | [c]CF50 | [d]CF50T | [e]CC200 | [f]CC200T |
|---|---|---|---|---|---|---|
| Sensitivity ($\Delta$OD/(Gy·cm)) | 0.0518 | 0.0315 | 0.0413 | 0.0549 | 0.0739 | 0.1010 |
| Sensitivity enhancement | - | -39.2% | - | +32.7% | - | +36.6% |
| $R^2$ value | 0.9960 | 0.9969 | 0.9954 | 0.9961 | 0.9995 | 0.9996 |

[a]CF30 = Clear Flex™ 30.

[b]CF30T = CF30 adding tartrazine.

[c]CF50 = Clear Flex™ 50.

[d]CF50T = CF50 adding tartrazine.

[e]CC200 = Crystal Clear™ 200.

[f]CC200T = CC200 adding tartrazine

*et al.* evaluated several formulations of PRESAGE with different Shore hardness [39]. Deformable (i.e., very low Shore hardness) PRESAGE formulations exhibited lower sensitivity than the non-elastic polyurethane matrix (i.e., Shore hardness 80 D). In terms of oxygen diffusivity, a soft material has a much greater permeability to small molecules than a hard material [40]. A higher oxygen concentration could occur in soft polyurethane and a high oxygen diffusivity reduces the dosimetric response. Alqathami *et al.* studied the potential influence of oxygen on the efficiency of the PRESAGE dosimeter [41]. A noticeable oxygen influence on the sensitivity of PRESAGE has been observed, and there was no influence on the $R^2$ value, absorption spectra, or stability of PRESAGE. In this study, the higher-hardness formulation with 80 D and 50 A showed more significant sensitivity enhancement than the lower-hardness formulations (30 A). By incorporating tartrazine, the dosimetric characteristics were further improved for the formulations with 50 A and 80 D. However, for the formulation with 30 A, the characteristics were degraded by incorporating tartrazine. Tartrazine is a synthetic lemon-yellow azo-dye used as a food coloring agent [42]. This azo dye is capable of producing free radicals. Several

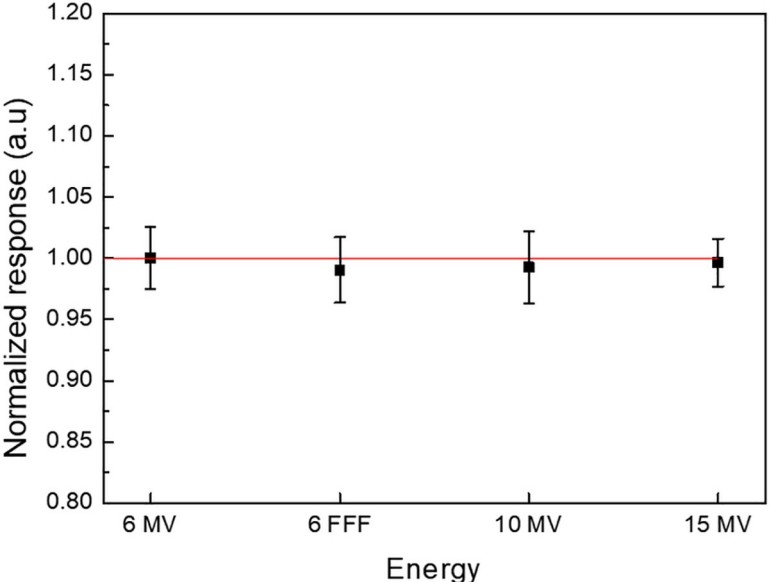

**Fig 6. Energy dependence of CC200T.**

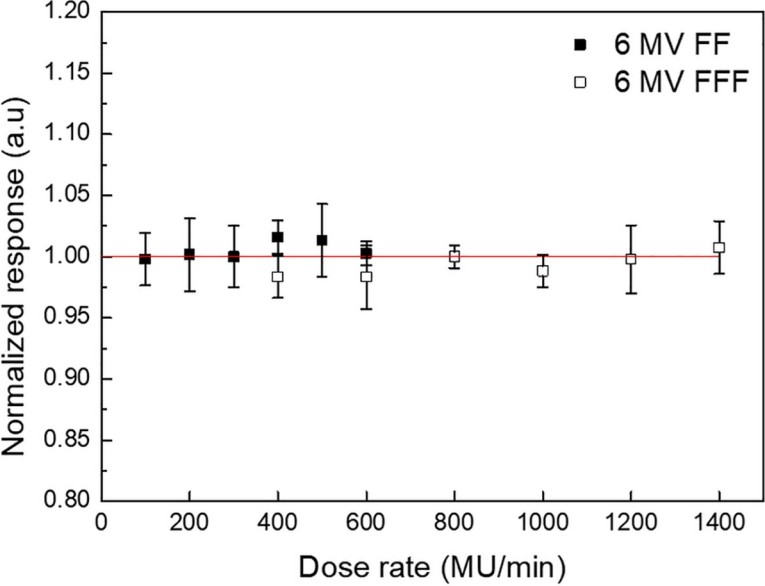

**Fig 7. Dose rate dependence of CC200T.**

biology studies have reported the effect of tartrazine [43–45]. Tartrazine has also been investigated as a gamma radiation dosimeter, and the results showed that tartrazine degradation was detected with increasing gamma dosage [46]. According to this result, tartrazine could increase the production of free radicals in higher-hardness formulations with 50 A and 80 D; thus, the sensitivity was enhanced significantly and there was no influence on $R^2$ value in our study. However, the reaction of free radicals could be reduced by oxygen diffusivity in soft polyurethane despite incorporating tartrazine. Thus, the sensitivity decreased significantly in the low-hardness formulation with 30 A in our study. It is important to note that tartrazine has a significant effect on the dosimetric characteristics of PRESAGE dosimeters using a rigid polyurethane resin.

The dosimeter fabricated with CC200T formulation showed negligible energy dependence and dose rate dependence and demonstrated enhanced sensitivity. However, since there was a limit to the production of large size dosimeters in our laboratory, further studies are needed to manufacture the large size dosimeter fabricated with CC200T formulation and perform IMRT QA to verify that the dosimeter is suitable for clinical use.

## Conclusions

We investigated the influence of tartrazine on the dose response characteristics of PRESAGE dosimeters with various polyurethanes using absorbance and optical density acquired by spectrometer and optical CT sacanner. By incorporating tartrazine, the sensitivity was significantly increased for two kinds of polyurethane resins: Clear Flex 50 (Shore hardness 50 A) and Crystal Clear 200 (Shore hardness 80 D). This high sensitivity dosimeter can be applied to perform the 3D dose QA for IMRT or VMAT.

## Author Contributions

**Conceptualization:** Hong-Gyun Wu, Jung-in Kim.

**Data curation:** Jin Dong Cho, Jaeman Son.

**Formal analysis:** Jaeman Son, Chang Heon Choi.

**Funding acquisition:** Hong-Gyun Wu, Jung-in Kim.

**Investigation:** Jin Sung Kim, Jong Min Park.

**Project administration:** Hong-Gyun Wu.

**Supervision:** Hong-Gyun Wu, Jong Min Park, Jung-in Kim.

**Validation:** Jin Sung Kim, Jong Min Park.

**Visualization:** Chang Heon Choi.

**Writing – original draft:** Jung-in Kim.

**Writing – review & editing:** Jin Dong Cho.

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
