## [Decision Letter · Decision Letter 0]

17 Jan 2020

PONE-D-19-33935

Improvement in sensitivity of radiochromic 3D dosimeter based on rigid polyurethane resin by incorporating tartrazine

PLOS ONE

Dear Dr. Kim,

Thank you for submitting your manuscript to PLOS ONE. After careful consideration, we feel that it has merit but does not fully meet PLOS ONE’s publication criteria as it currently stands. Therefore, we invite you to submit a revised version of the manuscript that addresses the points raised during the review process.

We would appreciate receiving your revised manuscript by Mar 02 2020 11:59PM. To enhance the reproducibility of your results, we recommend that if applicable you deposit your laboratory protocols in protocols.io, where a protocol can be assigned its own identifier (DOI) such that it can be cited independently in the future. For instructions see: http://journals.plos.org/plosone/s/submission-guidelines#loc-laboratory-protocols

We look forward to receiving your revised manuscript.

Kind regards,

Kalisadhan Mukherjee

Academic Editor

PLOS ONE

Journal Requirements:

http://www.elimpex.com/new/products/radiation_therapy/Gafchromic/content/GafChromic_EBT-2_20101007.pdf

In your revision ensure you cite all your sources (including your own works), and quote or rephrase any duplicated text outside the methods section. Further consideration is dependent on these concerns being addressed.

Additional Editor Comments (if provided):

We have received the comments from 2 reviewers. As mentioned by the reviewers, the scientific rigor of the manuscript is poor in it's current form. Authors are suggested to consider each comments of the reviewers critically to improve the quality of the manuscript. Include the scientific insights in the manuscript. Please improve the English and grammar as well. Major revision is recommended.

Reviewers' comments:

Reviewer's Responses to Questions

**Comments to the Author**

1. Is the manuscript technically sound, and do the data support the conclusions?

Reviewer #1: No

Reviewer #2: Yes

2. Has the statistical analysis been performed appropriately and rigorously? 

Reviewer #1: No

Reviewer #2: N/A

3. Have the authors made all data underlying the findings in their manuscript fully available?

Reviewer #1: Yes

Reviewer #2: Yes

4. Is the manuscript presented in an intelligible fashion and written in standard English?

Reviewer #1: Yes

Reviewer #2: Yes

5. Review Comments to the Author

Reviewer #1: The authors presented a study on the improvement of sensitivity of PRESAGE radiochromic 3D dosimeters, obtained by adding tartrazine to different polyurethane resins. The importance of 3D dosimetry in medical applications and the wide use of radiochromic dosimeters make this topic interesting and worthy of being studied. However, the results presented in this manuscript are not reported with the proper scientific rigor. The manuscript in its current form is too confused, in particular in the description of the experimental methods and in the presentation of results. Some parts are very hard to read, by making difficult the review process. I think that this manuscript is not ready for publication in PLOS ONE and it needs an overall review of its general framework. Nevertheless, the re-submission is encouraged once these points will be properly addressed. My major amendments are listed below.

Introduction

Line 61. In the clinical practice, 3D dosimetry is mainly performed with dosimeters other than chemical ones, for example by moving ionization chambers or silicone diodes or TLDs etc. in different positions of e.g. a water phantom (take a look at the following publication: https://www.ncbi.nlm.nih.gov/pmc/articles/PMC5368627/). The statement al line 61 should be rearranged.

Line 63. This statement is not clear since optical absorption, scattering, X-ray absorption, NMR are not properties of dosimeters, but physical phenomena.

Line 78. “This change in optical density (OD) is linear with respect to the absorbed dose up to 100 Gy”. This statement is uncomplete since it is important to specify the full range (linear in the range…).

Lines 93-99. This paragraph is not well linked to the rest of the introduction. Additional sentence(s) aiming at introducing the reason for discussing EBT and EBT2 films are needed. Furthermore, the sentences of lines 96-98 must be better explained; the conclusions are faint and should be supported by more well-founded arguments.

Lines 100-102. Also this paragraph needs a better introduction. For example, it is not said why the authors choose to add just the tartrazine as a yellow dye. What is its chemical structure, are there similarities with the dye of EBT films ? …

Materials and Methods

Line 106. “Radiochromic plastic gels were fabricated…”. Did the authors fabricated commercial PRESAGE dosimeters with the addition of tartrazine or they developed new dosimeters?

Line 157. Was the uniformity of the radiation field measured/calculated? The uniformity is generally expressed with a percentage.

Line 161. Monitor Unit is not physical quantity. Gy/min and its multiples/submultiples are the correct units.

Lines 172-173. From this sentence, I understand that PRESAGE dosimeters change the absorbance for exposure to VIS light. Since this aspect is crucial for the use of PRESAGE dosimeters in the practice, a comment is needed.

Line 179. Radiochromic films are sensitive to UV light, namely they show an increasing darkness with the increase of the exposure to UV light. I expect a same behaviour for PRESAGE dosimeters. Since the Xe flash lamp emits in the range 200-830 nm, could this be an issue for this work? Can the authors comment on this aspect?

Lines 199-207. It is not clear which is the actual output quantity of the spectrometer (Absorbance, Net absorbance?); how the other quantities are evaluated and how the spatial information (delta Absorbance/Gy*cm) in the dose sensitivity is evaluated. Optical density and absorbance are usually used to describe the same quantity, while in this manuscript they seem to indicate different quantities (“The dose response curve was obtained by plotting optical density and maximum absorbance as a linear function of absorbed dose.” and “using optical absorbance and optical density measurements” in the conclusions). These points are crucial for the understanding of all the manuscript and need to be rewritten.

Results

Lines 223-227. This presentation of results (Zeff for 2 formulations and presence of some elements in the compound) is meaningless. It would be interesting to see the results (tables, plots etc.) of XRF measurements for all (or at least for some) formulations. Moreover, it is said the chemical composition includes C, H, O, N and Br. However, no results are given for the concentration of these elements. This is a useful information for possible further studies on these dosimeters.

Line 231. “Very stable” is meaningless. Can the authors quantify how the spectra are stable after irradiation?

Line 232. How the results are consistent with OD measurements? Are OD measurements those performed with CT scanning (see comments at line 199-207)?

Lines 232-242. Scientific results should be presented in plots or tables which make clear the text.

Table 2. Express correctly the significant figures of the quantity “sensitivity at lambda max” in table 2.

Lines 299-301. This sentence is not clear.

Discussion

Also in this case, the reading is very hard. It is not clear which is the point. A synthesis will help.

Conclusion

What is the sensitivity enhancement of the most performing fabricated dosimeter compared to the commercial PRESAGE dosimeter? It would be useful to report this information in the conclusion.

Reviewer #2: General comments

This is an interesting contribution that clearly demonstrated the advantage of adding tartrazine to a gafchromic 3D dosimeter for higher radiation sensitivity.

Major comments

1. Could you explain the physical mechanism for the increase of absorption by adding tartrazine? Yellow dye was added to EBT2 to make it less sensitive to room light as you mention in the introduction.

2. You need to mention why one needs a 3D dosimeter with higher radiation sensitivity in the Introduction.

3. The effective Z of the dosimeter material was determined to be 10.5, which is much higher than that of water, which is about 7.4 at the X-ray energy used for imaging such as CT. Are you concerned about the high Z-effective as a dosimeter used in radiation oncology?

4. XRF provides us atomic compositions. How did you calculate Z-eff? What formula did you use? What photon energy did you use to calculate it?

5. The discussion on EBT films in lines 337-365 should be shortened. Only a reason it is here is to give a rationale for using a yellow dye with a PRESAGE type dosimeter.

6. PLOS authors have the option to publish the peer review history of their article (what does this mean?). If published, this will include your full peer review and any attached files.

Reviewer #1: No

Reviewer #2: No

---

## [Author Response · Author response to Decision Letter 0]

18 Feb 2020

We appreciate the editor/referee’s time and effort in reviewing this work. We revised our manuscript following reviewer’s points. Thank you for your consideration.

---

## [Decision Letter · Decision Letter 1]

2 Mar 2020

Improvement in sensitivity of radiochromic 3D dosimeter based on rigid polyurethane resin by incorporating tartrazine

PONE-D-19-33935R1

Dear Dr. Kim,

We are pleased to inform you that your manuscript has been judged scientifically suitable for publication and will be formally accepted for publication once it complies with all outstanding technical requirements.

With kind regards,

Kalisadhan Mukherjee

Academic Editor

PLOS ONE

Additional Editor Comments (optional):

Reviewer has made comment in favor of the manuscript. Thus it can be accepted for publication.

Reviewers' comments:

Reviewer's Responses to Questions

**Comments to the Author**

1. If the authors have adequately addressed your comments raised in a previous round of review and you feel that this manuscript is now acceptable for publication, you may indicate that here to bypass the “Comments to the Author” section, enter your conflict of interest statement in the “Confidential to Editor” section, and submit your "Accept" recommendation.

Reviewer #2: All comments have been addressed

2. Is the manuscript technically sound, and do the data support the conclusions?

Reviewer #2: Yes

3. Has the statistical analysis been performed appropriately and rigorously? 

Reviewer #2: N/A

4. Have the authors made all data underlying the findings in their manuscript fully available?

Reviewer #2: Yes

5. Is the manuscript presented in an intelligible fashion and written in standard English?

Reviewer #2: Yes

6. Review Comments to the Author

Reviewer #2: Thank you very much for clarifying my questions in your rebuttals and the revisions. The study is interesting and should be published.

7. PLOS authors have the option to publish the peer review history of their article (what does this mean?). If published, this will include your full peer review and any attached files.

Reviewer #2: No

---

## [Editor Report · Acceptance letter]

4 Mar 2020

PONE-D-19-33935R1 

Improvement in sensitivity of radiochromic 3D dosimeter based on rigid polyurethane resin by incorporating tartrazine 

Dear Dr. Kim:

I am pleased to inform you that your manuscript has been deemed suitable for publication in PLOS ONE. Congratulations! Your manuscript is now with our production department. 

With kind regards,

on behalf of

Dr. Kalisadhan Mukherjee 

Academic Editor

PLOS ONE